

# Times and partners are a-changin': relationships between declining food abundance, breeding success, and divorce in a monogamous seabird species

David Pelletier[1,2] and Magella Guillemette[2]

[1] Département de Biologie, Cégep de Rimouski, Rimouski, Québec, Canada
[2] Département de Biologie, Chimie et Géographie, Université du Québec à Rimouski, Rimouski, Québec, Canada

Corresponding author
David Pelletier,
david.pelletier@cegep-rimouski.qc.ca

## ABSTRACT

Seabirds exhibit considerable adjustment capacity to cope with environmental changes during the breeding season and to maximize lifetime reproductive output. For example, divorce has been proposed to be an adaptive behavioral strategy in social monogamous species, as a response to poor conditions and low breeding success. Here, we studied divorce at the population and individual levels in northern gannets (*Morus bassanus*, hereafter gannets) nesting on Bonaventure island (Quebec, Canada). At the population level, we used Granger's method for detecting and quantifying temporal causality between time series (from 2009 to 2019) of divorce rate and breeding success of gannets ($n = 809$) and we evaluated the relationship between breeding success and biomass of their two principal prey (Atlantic mackerel, *Scomber scombrus*, and Atlantic herring, *Clupea harengus*). Our results indicated that breeding success is mainly influenced by the spawning-stock biomass of Atlantic mackerel, and a decrease in breeding success is followed by an increase in divorce rate with a 1-year lag. However, the effect of the interaction between breeding success and year on the proportion of individuals that divorced showed significant inter-annual variation. At the individual level, our results support the adaptive strategy hypothesis of divorce. Indeed, gannets that changed partners did so following a reproductive failure, and there was an increase in breeding success 1 year following the divorce. Being central place foragers, opportunities for dispersal and adaptation are often limited for breeding seabirds in a context of low food abundance. We suggest that behavioral flexibility expressed as divorce would be an efficient short-term strategy for maintaining reproductive performance.

## INTRODUCTION

Seabirds demonstrate considerable ecological, demographic, life-history and behavioral adjustment capacity (*e.g.*, *Garthe, Camphuysen & Furness, 1996*; *Hamer et al., 2007*; *Weimerskirch, 2002*) to respond and adjust to short- and long-term changes in ocean

conditions (*e.g.*, *Aebischer, Coulson & Colebrookl, 1990*; *Sydeman et al., 2009*). They are resilient to environmental change, *i.e.*, they have the ability to survive and recover from a perturbation (*Williams et al., 2008*). For example, under conditions of low prey availability, parents may modify their attendance and foraging behavior (*Cairns, 1987, 1992*; *Piatt et al., 2007*). Normally, both parents divide care to protect and feed their chicks (*Schreiber & Burger, 2002*). When food is scarce, they are more likely to leave a younger chick earlier to obtain sufficient food (*Regehr & Montevecchi, 1997*). They can increase time spent foraging and the distance traveled to find prey (*Guillemette et al., 2018*). However, such behavioral adjustments in colonial breeders may compromise their breeding success as they must leave their offspring temporarily unattended. For example, chicks left unprotected by their parents may then be attacked by adults from nearby sites (*Ashbrook et al., 2008*), or to be assaulted by non-breeders attempting to usurp sites (*Porter, Anderson & Ferree, 2004*; *Hamer et al., 2007*), or killed by predators (*Oro & Furness, 2002*). When food conditions are poor, the usual benefit of high-density breeding as protection from predators may diminish and thus reduce breeding success (*Danchin & Wagner, 1997*). Poor breeding performance may even destabilize pair bonds in monogamous seabirds species (*Ens, Choudhury & Black, 1996*; *Bried & Jouventin, 2002*; *Dubois & Cézilly, 2002* but see *Choudhury, 1995*; *Taborsky & Taborsky, 1999*).

Birds exhibit a diversity of mating systems, with various degrees of fidelity to a partner according to the duration of the relationship, from continuous partnership with no promiscuity to social monogamy with limited genetic exchange (*Clutton-Brock, 1991*; *Black, 1996*). Divorce has been recorded in 92% of socially monogamous bird species (*Jeschke & Kokko, 2008*), with divorce rates being highly variable both among and within species (*Black, 1996*). Divorce rate is particularly variable among species, ranging from partners repairing every breeding season (100% divorce, *e.g.*, imperial shag (*Leucocarbo atriceps*), great blue heron (*Ardea herodias*) (*Jeschke & Kokko, 2008*) to strict partner fidelity (0% divorce, *e.g.*, Buller's albatross (*Thalassarche bulleri*) (*Ens, Choudhury & Black, 1996*), south polar skua (*Catharacta maccormicki*), *Mercier, Yoccoz & Descamps, 2021*)). Divorce rates may also vary considerably between populations of the same species (*e.g.*, 8 to 85% in the blue tit (*Parus caeruleus*)) (*Dhondt & Adriaensen, 1994*) and between years (*e.g.*, from 13% to 50% in black-legged kittiwake (*Rissa tridactyla*) (*Mercier, Yoccoz & Descamps, 2021*). The causes of this variation in divorce rates are not yet fully explained, but it has recently been shown that the probability of divorce is directly affected by environmental variability in seabirds (*Ventura et al., 2021*).

Various hypotheses have been proposed concerning the costs and benefits associated with mate retention *vs*. divorce (*Choudhury, 1995*). Because biparental care (and the related social monogamy) is crucial in most monogamous bird species (*Bennett & Owens, 2002*), divorce represents a way to improve potentially problematic partnerships that may result from different types of factors implied in the initial mate choice (*Johnston & Ryder, 1987*; *Moller, 1992*; *Sullivan, 1994*; *Choudhury, 1995*; *Botero & Rubenstein, 2012*), and/or to obtain more genetically diverse offspring (see references in *Arnqvist & Kirkpatrick, 2005*). Two main groups of hypotheses have been proposed to explain divorce: an adaptive strategy that increases the breeding success of at least one of the two partners; a

random event that results from another strategy (see review by *Choudhury, 1995*). For divorce to be an adaptive strategy, breeding success post-divorce should be higher than pre-divorce (*Choudhury, 1995*; *Black, 1996*; *Dubois & Cézilly, 2002*). Thus, divorce may be seen as a response to unfavorable environmental and breeding conditions or to the possibility of more favorable conditions with a partner of improved quality. For example, if the divorce is the result of a random or unintended effect of another process, it should not be related to the reproductive success of the previous breeding season. Thus, divorce would potentially lead to a decrease or no change in breeding success.

Most of the studies of divorce focus on the general patterns observed at the population level, but these rarely provide insights into processes happening at the individual level and at the finer timescale (*i.e.*, between two breeding seasons). For example, the 'success-stay/failure-leave' pattern (*Schmidt, 2004*) is often observed at the nest or population level, where those pairs that fail divorce, and those that succeed stay together. However, some pairs stay together after a failure, while others divorce after a successful breeding attempt (*e.g.*, *Brooke, 1978* for manx shearwater, *Puffinus puffinus*, *Harris et al., 1987* for oystercatcher, *Jones & Montgomerie, 1991* for least auklet, *Aethia pusilla*, *Ramsay et al., 2000* for black-capped chickadees, *Poecile atricapillus*, *Saino et al., 2002* for barn swallow, *Hirundo rustica*).

The northern gannet (*Morus bassanus*, hereafter gannets) breed in large colonies and they are philopatric to their breeding site (*Nelson, 2002*). This species is a long-lived plunge-diving predator, highly territorial and aggressive at its nest site and has a wide breeding distribution in the North Atlantic. This species is reported as socially monogamous with the suggestion of mate retention for life with a 17% divorce rate reported (*Nelson, 2002*). Gannets exploit cold, nutrient-rich waters, and rely on seasonally abundant fish stocks including Atlantic mackerel (*Scomber scombrus*), Atlantic herring (*Clupea harengus*), capelin (*Mallotus villosus*) and sand lance (*Ammodytes* sp.) (*Garthe et al., 2007*; *Guillemette et al., 2018*). However, unlike other populations that appear to be increasing, gannets breeding in the Gulf of St. Lawrence show a decline in breeding success to levels even lower than in the 1960s (around 30%, while DDE contamination rate was very high). Recently, from data spanning 35 years, breeding success of this species was positively related to mackerel abundance (*Guillemette et al., 2018*). High nest-site fidelity, low breeding success, gregarious behavior, and tolerance towards disturbance make this species a very good model for the study of divorce.

In this paper, we studied the occurrence of divorce at population and individual levels. **When do gannets divorce?** Our first objective was to investigate, at the population level, the relationship between prey abundance, diet and breeding success of gannets followed by the relationship between breeding success and partnership status of gannets. Our first hypothesis was that divorce is influenced by low breeding success and triggered by decreasing prey abundance in the marine ecosystem. We predicted that an ecosystem-wide decrease in the biomass of key gannet prey species is associated with decreased breeding success at the population level (P1). We also predicted that a decrease in mackerel (the preferred prey) in an increasingly heterogeneous diet is related to a decrease in breeding success (P2). Our third prediction was that low breeding success is followed by an increase
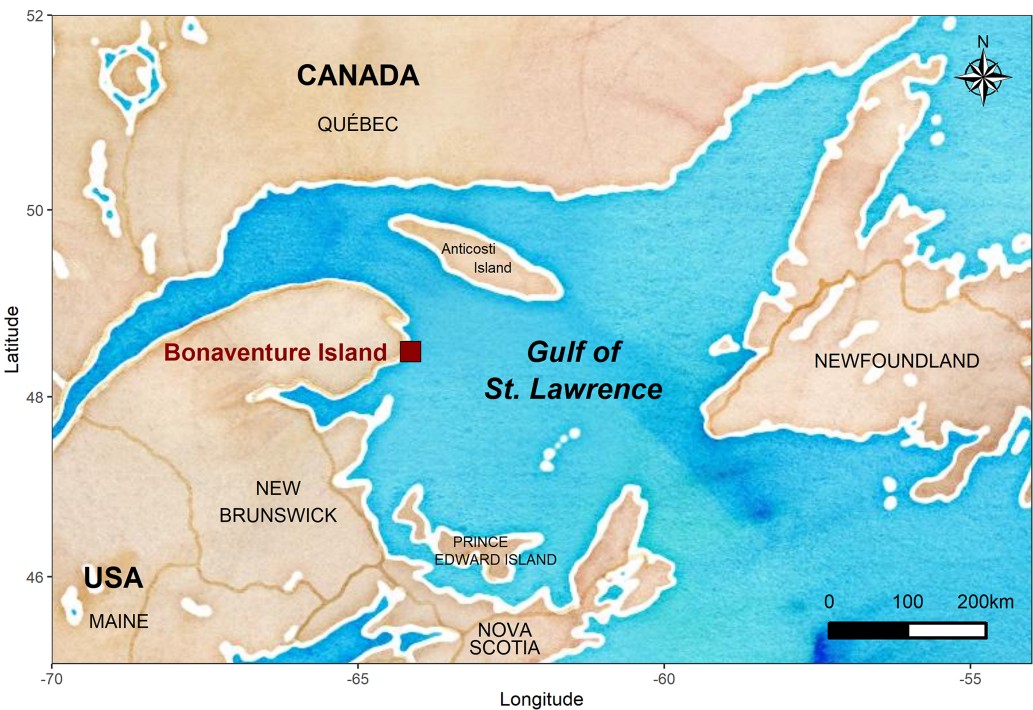

**Figure 1 Location of the Bonaventure Island's northern gannet colony.** The map was generated in R (*R Core Team, 2020*) using the 'ggmap' (*Kahle & Wickham, 2013*) and 'ggplot2' (*Wickham, 2016*) packages with map tiles by Stamen Design (www.stamen.com) and data by OpenStreetMap, under ODbL, under CC BY 3.0 (www.creativecommons.org).

in divorce rate (P3). **Why do gannets divorce?** Our second objective was to test the hypothesis, at the individual level, that divorce is an adaptive strategy. We thus predicted that gannets would tend to divorce more frequently following breeding failure (P4) and that divorced individuals would increase their breeding success after mate change (P5).

# METHODS

## Study site and field work

The fieldwork was conducted on Bonaventure Island (48°30′ N, 64°09′ W) located in the Ile-Bonaventure-et-du-Rocher-Percé National Park in the Gulf of St. Lawrence (GSL, Quebec, Canada) where a long-term study started in 2008 (Fig. 1). This colony holds approximately 50,000 pairs of breeding gannets as well as several thousand immatures. Annually, this colony is monitored for partnership status and breeding success from incubation to second part of the chick-rearing period (from May to September). Gannets were caught from 108 to 184 nests per year using a noose-pole within four plots in the peripheral section of the colony (the first 108 nests were monitored through 2019 and we added nests over the years). All bird capture and handling methods were approved by the Animal Care Committee (ACC) of the Université du Québec à Rimouski (CPA-49-12-102, CPA-65-16-177), and complied with the guidelines of the Canadian Council on Animal Care (CCAC). Field experiments were approved by Canadian Wildlife Service-Environment and Climate Change Canada (permit numbers SC25, RE-27) and by

Societe des establissements de plein air du Quebec (permit numbers PNIBRP-2008-001 to PNIBRP-2019-001). Birds were marked with a U.S. Fish and Wildlife Service steel ring and an alphanumeric coded and colored plastic band (permit number 10704). Because gannets regurgitate regularly during handling, we were able to study the diet annually throughout the breeding season. We were therefore able to identify the main prey items and establish their proportion in their diet (see *Guillemette et al. (2018)*). Occurrences of individually banded gannets within the colony were established visually twice daily over a period from the end of May (before hatching) to the end of August (or September), from 2008 to 2019. We identified established partnerships in the set of monitored nests coupled with an intensive visual survey of the study area and surrounding sectors to ensure that we resighted all live breeding birds. Because gannets have a strong breeding philopatry (*Nelson, 2002*), the absence of a partner within the colony was considered as a death or a skipping reproductive behavior.

## Sex determination

Four thoracic covert feathers were plucked during capture and stored in plastic bags at −20 °C until subsequent analysis. DNA was isolated from these feathers. Quills were cut into 2- to 5-mm-long pieces and submerged in 50 μL of a solution of QuickExtract$^{TM}$ DNA Extraction Solution (Lucigen, Middleton, WI, USA) in a 96-wells plate. Each plate was incubated in a G-Storm GS4 thermal cycler during 5 h at 65 °C for DNA extraction, and 10 min at 95 °C for ending the process. One set of primers was used for the amplification of CHD gene (Chromo Helicase DNA-binding gene): P2 (5′-TCTGCATCGCTAAAT CCTTT-3′) and P8 (5′-CTCCCAAGGATGAGRAAYTG-3′) described by *Griffiths et al. (1998)* for sex determination in birds. The amplification was carried out in a total reaction volume of 25 μL containing 1× ThermoPol buffer (New England Biolabs, Ipswich, MA, USA), 0.2 mM of each dNTP, 0.2 μM of each primer (P2 and P8), 0.05 units of *Taq* polymerase, 5 uL of DNA (100 ng), and HPLC grade water for completing volume. PCR was performed in a G-Storm GS4 thermal cycler. An initial denaturing step at 94 °C for 1 min 30 s was followed by 40 cycles of 48 °C for 45 s, 72 °C for 45 s and 94 °C for 30 s. A final run of 48 °C for 1 min and 72 °C for 5 min completed the program. PCR products were separated by electrophoresis for 3 h at 95 V in a 3% agarose gel stained with 1× GelRed Nucleic Acid Gel Stain (Biotium, San Francisco, CA, USA). One μL of the DNA amplified solution was diluted in 4 μL of HPLC grade water and mixed with 1 μL of 6× gel loading buffer composed of 0.03% Bromophenol Blue, 0.03% Xylene Cyanol FF, 60 mM EDTA, pH 7.6 and 60% glycerol in HPLC grade water (*Griffiths et al., 1998*; *Redman et al., 2002*).

## Mackerel and herring biomass
### Atlantic mackerel

Atlantic mackerel (*Scomber scombrus*) is a schooling pelagic species. Its biomass was estimated and provided by the Ecosystems and Oceans Science service at the Department of Fisheries and Oceans Canada (DFO) (*Smith et al., 2020*). These data come from an estimation model produced during an analytical assessment of the population dynamics

integrating fisheries-independent data (biomass index) and fisheries-dependent data (landings and catch-at-age). Biomass index is an estimate of the spawning stock biomass of mackerel deriving from the annual standardised ichtyoplankton surveys according to the total egg production method (TEPM) (*Saville, 1977*). These surveys have taken place in the Southern GSL since 1979 and assume that this region is the main mackerel spawning area in Canadian waters, as exploratory surveys elsewhere found little to no mackerel eggs (see *Smith et al., 2020*). Consequently, this mackerel biomass estimate is considered as an estimate of mackerel abundance inside and outside Gulf of St. Lawrence (NAFO subareas 3 and 4, including eastern coast of Newfoundland, southern coast of Nova Scotia). Consistent with the fact that fish found in the regurgitation of gannets are up to 42 cm long and mackerel under 10 years have a lower average length-at-age (*Smith et al., 2020*), all age classes of mackerel biomass were retained. Then, from the estimation model, we calculated the total stock biomass (TSB) given $TSB$: $N_{a,y} * W_{a,y}$, where $N_{a,y}$ is the number of fish at age $a$ in a given year $y$ in the population at January first estimated by the model, and $W_{a,y}$ is the weight-at-age $a$ in a given year $y$. Data from 1979 to 2019 (updated in 2020) were used in our study.

### *Atlantic herring*

Biomass estimates of the gannet's second principal prey (*Clupea harengus*) were also provided by the DFO, but the assessment process is different. Herring population in the GSL consists of two genetically distinct spawning components: spring spawners and fall spawners. Both stocks are managed separately for assessment and fisheries (*DFO, 2018*, *2019*, *2020*). Herring stocks are managed in three NAFO Divisions of the GSL, but only data from the region mostly used by gannets (4T: Southern GSL) were used in our study (*DFO, 2020*). For the spring spawning component, a statistical catch-at-age (SCA) model with time-varying natural mortality and time-varying catchability to the fixed gear fishery was used (qSCA model). For the fall spawning herring component in the division 4T, a regionally-disaggregated assessment model is used for the North, Middle and South regions, but we summed the total for each sub-region. The fall spawning herring component is assessed using two SCA models: a qSCA and a qmSCA models (including time varying natural mortality), but because the mackerel assessment and spring spawning herring components ignore natural mortality, qSCA data were used as biomass estimates of fall spawning herring component biomass. Data from 1979 to 2019 were used in our study (*DFO, 2020*).

### Quantification of diet, breeding success and divorce

We monitored diet of gannets from 2013 to 2019 by analyzing the content of the regurgitation they make when handled or disturbed. The proportions of occurrence of each prey species in the regurgitated food was calculated yearly and, from these proportions, we calculated the Shannon H-index as an index of diet diversity (where higher values signifying a more diverse diet, and inversely): $H = -\sum_{i=1}^{s}(Pi * \ln Pi)$ where $Pi$ = proportion of a species $i$ relative to the total number of species present in diet and $s$ = number of species encountered. We added data from 2004, 2005 and 2009 reported in

**Table 1 Total number of pairs of northern gannets monitored between 2009 and 2019 on Bonaventure Island colony and percentage of pairs with partners retained, lost or divorced per year.**

| Year $t$ | Number of monitored nests | Number of monitored pairs at year $t$ | Number of pairs with known partnership and breeding status at both years $t-1$ and $t$ | Number of pairs with PARTNER RETAINED | % | Number of pairs with PARTNER LOST | % | Number of pairs with PARTNER DIVORCED | % |
|---|---|---|---|---|---|---|---|---|---|
| 2009 | 108 | 73 | 0 | – | – | – | – | – | – |
| 2010 | 169 | 84 | 7 | 6 | 86 | 0 | 0 | 1 | 14 |
| 2011 | 173 | 88 | 12 | 9 | 75 | 1 | 8 | 2 | 17 |
| 2012 | 172 | 141 | 28 | 12 | 43 | 3 | 11 | 13 | 46 |
| 2013 | 180 | 111 | 54 | 26 | 48 | 7 | 13 | 21 | 39 |
| 2014 | 179 | 116 | 54 | 38 | 70 | 4 | 7 | 12 | 22 |
| 2015 | 180 | 125 | 62 | 50 | 81 | 3 | 5 | 9 | 15 |
| 2016 | 180 | 140 | 41 | 31 | 76 | 3 | 7 | 7 | 17 |
| 2017 | 181 | 142 | 60 | 42 | 70 | 8 | 13 | 10 | 17 |
| 2018 | 181 | 143 | 65 | 43 | 66 | 4 | 6 | 18 | 28 |
| 2019 | 184 | 146 | 80 | 63 | 79 | 6 | 8 | 11 | 14 |

*Guillemette et al. (2018)*. Given the high rate of mate changing within the monitored nests (108 to 184), breeding success and partnership status were studied in a total of 809 birds for a total of 704 pairs from 2008 to 2019 (Table 1). We estimated annual breeding success of the colony as the number of fledged chicks divided by the number of nests with eggs. A divorce is said to have occurred when two birds that bred together in year $t-1$ were alive and present in the colony at year $t$, but not breeding together. Mate retention occurred when both partners were together at year $t-1$ and $t$, and mate loss occurred when one of the two partners was absent in our study plot (*Coulson, 1966*; *Ens, Safriel & Harris, 1993*; *Choudhury, 1995*). Divorce rate is the number of divorced birds divided by the total number of birds alive.

## Statistical analyses

### Relationships between prey biomass and breeding success

As the relationship between prey biomass and breeding success are known to be nonlinear (*Cairns, 1987*; *Piatt & Sydeman, 2007*), we formulated generalized additive models (GAMs) with the 'gam' function of the 'mgcv' package (*Wood, 2022*). The annual breeding success of the northern gannets was modeled as a function of the biomass of their two main prey species: mackerel and herring (spring and fall populations). We used GAMs as they provide a flexible approach by not presuming a linear form of relationship and can be used to estimate nonlinear effects of independent variables and covariates on dependent variables (*Wood, 2017*). We wrapped independent variables (biomass) in the smooth function ('s()') and we fit our models with the restricted maximum likelihood (REML) method to get more reliable and stable results. Because data for prey biomass was available

until 1979 for prey biomass, we added breeding success data provided by Environment and Climate Change Canada and reported in *Guillemette et al. (2018)* (years added: 1979, 1984, 1989, 1994, 1999, 2004, 2005, 2008) to build these models. We tested for differences in annual means of prey biomass using analysis of variance (ANOVA) and *post hoc* Tukey HSD tests using the 'aov' and 'TukeyHSD' functions of the 'multcomp' package (*Hothorn & Westfall, 2008*). Normality was tested using the 'shapiro.test' function and homoscedasticity was tested using the 'leveneTest' function of 'car' package (*Fox & Weisberg, 2019*). When assumptions of normality and homoscedasticity failed, Kruskal–Wallis and associated multiple comparison tests were performed using 'kruskal. test' and the 'pairwise.wilcox.test' functions of the 'stats' package (*R Core Team, 2020*), respectively.

### Relationships between diet and breeding success

Correlations between gannet diet and breeding success were computed using Pearson's correlations. A bootstrap significance testing approach was applied to estimate the $P$-value of the correlation coefficients with the 'boot.pval()' function from the R package 'boot.pval' (*Thulin, 2021*). Bootstrap replicates were generated with the function boot() (R = 10,000, method = "pearson") in the R package 'boot' (*Canty & Ripley, 2017*).

### Granger-causality between breeding success and divorce rate

In a time series, values are often not independent of the preceding time points (*Borcard, Gillet & Legendre, 2018*). Due to this temporal dependency and strong autocorrelation, traditional analysis techniques are statistically inadequate for studying these relationships. From many methods used for detecting and quantifying causality between time series, Granger's method (*Granger, 1969*) is particularly suited for empirical investigations of cause-effect relationships in stochastic systems (*Eichler, 2012*). Granger-causality can be useful for detecting interactions between strongly synchronized variables in nonlinear or linear systems (*Sugihara et al., 2012*). According to *Granger (1969, 1988)*, causality evokes the two principles: (1) the effect does not precede its cause in time and (2) the causal series contains unique information about the series being caused that is not available otherwise (*Eichler, 2012*). As an example, a variable $X$ is said to "Granger-cause" $Y$ if the predictability of $Y$ (in some idealized model) declines when $X$ is removed from the universe of all possible causative variables, $U$, and the variable $Y$ (with a lag of 1 year or more) does not cause the variable $Y$. Granger-causality is not identical to causation in the classical sense, because it does demonstrate the likelihood of such causation (or the lack of such causation) more forcefully than simple contemporaneous correlation and it provides a framework that uses predictability as opposed to correlation to identify causation between time series variables (*Sugihara et al., 2012*). To test temporal Granger-causality between divorce rate and breeding success, the fundamental time series stationarity condition (no trend) was tested first to confirm that process has the property that the mean, variance and autocorrelation structure do not change over time. An autocorrelation function ('acf()') and an augmented Dickey-Fuller test (ADF) ('adf()' in 'vars' package in R) were used (*Pfaff, 2008*). The null hypothesis tested with the ADF test is that an observable series is

stationary, without negative or positive trend. For nonstationary series, we conducted analyses using residuals from the time linear regressions ($lm(Y \sim X)$) performed between variables $Y$ and $X$. Then, Granger-causality tests were implemented with vector autoregressive (VAR) process for bivariate time series (*i.e.*, with two time-dependent variables) comparisons (with 'VAR()' in 'causality' functions of 'vars' package) (*Pfaff, 2008*). The number of lags implemented in the test was determined with the 'VARselect()' function, according to the lowest value of Akaike information criterion for small sample (AICc) calculated in model comparisons performed with lags of 1, 2 and 3 years (*Burnham & Anderson, 2002*). Also, we compared these results with *cross-correlation function* (CCF) by measuring the similarity of the two series as a function of the displacement of one relative to the other (Fig. S1-Electronic Supplemental Material).

### Annual differences in divorce rate

At the individual level, to evaluate the effect of breeding failure or success on the divorce rate, we constructed generalized linear mixed models (GLMM) with divorce rate at year $t$ (where $t$ is the year of the partnership status determination) as dependent variable and breeding success at year $t - 1$ (success *vs.* failure) and year as independent variables (*Stroup, 2012*) (with 'lme4' package, *Bates et al. (2015)*). We used binomial distribution (with link = "logit"). To avoid pseudoreplication due to gannets sampled multiple years, individual was included as random effects.

### Differences in breeding success

At the individual level, to compare the breeding success between partnership status categories, we constructed GLMM to test differences in annual breeding success (dependent variable) between partnership status and time ($t$, $t - 1$, $t + 1$, where $t$ was the year of partnership status determination) (independent explanatory variables) (*Stroup, 2012*). We used binomial distribution (with link = "logit") because gannets lay only one egg and thus, breeding success (one chick or no chick per year) can be seen as a binary variable. Individual and year were included as random effects. Because sample sizes differed between categories of explanatory variables, pairwise comparisons were performed using estimated marginal means (from 'emmeans' package, *Lenth (2020)*) for each group, and *post hoc* pairwise comparisons adjusted by Tukey were applied to test group differences. There was no model simplification and all terms were retained in all the models above (confirmed with the lowest AIC).

All statistical analyses were performed using R version 4.0.3. (*R Core Team, 2020*) and plots were made using OriginPro Version 9.8.0.200 (*OriginLab Corporation, 2020*) or 'ggplot2' (*Wickham, 2016*). Results with $P < 0.05$ were considered significant. Values given in text are mean ± standard error of the mean (S.E.M.).

## RESULTS

The biomass of the principal prey of the northern gannets showed substantial interannual variability in the Gulf of St. Lawrence between 1979 and 2009, just like their diet, breeding success, and divorce rate. Between 1979 and 2019, coefficients of variation (CV) calculated for mackerel and herring biomass (spring and fall populations) and gannets breeding

**Table 2 Summary statistics for prey biomass, diet, breeding success and divorce rate of northern gannets: biomass of Atlantic mackerel and Atlantic herring in the Gulf of St. Lawrence between 1979 and 2008 ($n$ = 8 years), and 2009 and 2019 ($n$ = 11 years); proportion of mackerel and herring in diet, and Shannon H-index calculated to evaluate diet diversity (from 2004 to 2019, $n$ = 10); breeding success between 1979 and 2008 ($n$ = 8) and between 2009 and 2019 ($n$ = 11); and divorce rate between 2009 and 2019 ($n$ = 11 years).**

| | | Parameter | Mean ± s.e.m. | Range | Year$_{min}$ | Year$_{max}$ |
|---|---|---|---|---|---|---|
| PREY BIOMASS | 1979–2008 | Mackerel biomass (t) | 274,100 ± 42,751 | 110,402–475,581 | 1999 | 1989 |
| | | Herring spring biomass (t) | 74,605 ± 15,906 | 23,476–143,636 | 2004 | 1994 |
| | | Herring fall biomass (t) | 216,972 ± 34,944 | 52,189–359,058 | 1979 | 2008 |
| | 2009–2019 | Mackerel biomass (t) | 67,336 ± 8,623 | 46,183–141,243 | 2014 | 2009 |
| | | Herring spring biomass (t) | 38,351 ± 1,447 | 29,476–45,228 | 2016 | 2010 |
| | | Herring fall biomass (t) | 395,529 ± 38,855 | 174,161–558,909 | 2019 | 2011 |
| DIET OF GANNETS | 2004–2019 | Proportion of mackerel | 0.43 ± 0.05 | 0.25–0.70 | 2013 | 2005 |
| | | Proportion of herring | 0.18 ± 0.04 | 0.01–0.45 | 2019 | 2004 |
| | | Diet diversity (Shannon index) | 1.29 ± 0.10 | 0.66–1.73 | 2005 | 2019 |
| 1979–2008: Breeding success (chick.yr$^{-1}$) | | | 0.70 ± 0.02 | 0.61–0.75 | 2008 | 1984 |
| 2009–2019: Breeding success (chick.yr$^{-1}$) | | | 0.32 ± 0.05 | 0.03–0.60 | 2012 | 2018 |
| 2009–2019: Divorce rate (divorced.survival$^{-1}$) | | | 0.22 ± 0.03 | 0.13–0.46 | 2011 | 2012 |

success were 85%, 63%, 45%, and 49%, respectively (Table 2 Between 2004 and 2019, the gannet's diet also showed great variability as well as an increase in species diversity in the last years (Table 2). Between 2009 and 2019; CV calculated for divorce rate and breeding success of gannets and for mackerel and herring biomass (spring and fall populations) were 53%, 52%, 42%, 13%, and 33%, respectively.

Between 1979 and 2019, the abundance of the two main prey of gannets was very different (Kruskall–Wallis (KW) test, $\chi^2$ = 30.0, df = 2, $P$ < 0.0001). However, from 1979 to 2008, mackerel biomass and fall component of herring biomass were similar ($P$ = 0.50), and spring component of herring biomass was 2.5 to 2.9-fold lower ($P$ = 0.007, Table 2). From 2009 to 2019, the biomass of mackerel and the spring component of herring dropped dramatically while the fall component of herring continued to increase. Of the three groups, the fall component of herring was the dominant species (10-fold higher than the other two, $P$ < 0.0001). Despite this inferiority in the ecosystem, mackerel biomass estimates accounted for the whole Northwest Atlantic (NAFO subareas 3 and 4) while herring biomass estimates (spring and fall) covered a much smaller area, only the southern part of the Gulf of St. Lawrence (NAFO division 4T). Moreover, despite its lower biomass, Atlantic mackerel was the dominant species in the gannet diet from 2004 to 2019, ranging from 25% to 70% (KW test, $\chi^2$ = 16.3, df = 3, $P$ = 0.0001).

From 2009 to 2019, the breeding success of the northern gannet in our study has decreased by 2.2 times compared to the previous three decades (1979–2008, Table 2, KW test, $\chi^2$ = 13.2, df = 1, $P$ = 0.0003). During the same period, one in five gannets changed partners annually, rising to almost one in two in 2012, the worst breeding success result recorded in our study (3%).

**Table 3 Generalized additive models between the breeding success (BrS) of northern gannets and the smoothed function of biomass of Atlantic mackerel (s(Mack)) and Atlantic herring (spring population: s(HerrS) and fall population: s(HerrF)).** The best parsimonious model (in bold) had only significant variables and was chosen using the highest proportion of variation explained by the model ($R^2$ adjusted), the highest percentage of deviance explained (DEV) and the smallest Akaike index criterion for small sample (AICc). Significant variables in models are represented by asterisks (*$P < 0.05$, **$P < 0.01$).

| Models | $R^2_{adj.}$ | DEV (%) | $AIC_c$ | Delta | Weight |
|---|---|---|---|---|---|
| **BrS ~ s(Mack)**** | **0.56** | **61** | **167.2** | **0.00** | **0.39** |
| BrS ~ s(HerrF) + s(Mack)** | 0.69 | 76 | 167.3 | 0.10 | 0.37 |
| BrS ~ s(HerrF)** | 0.39 | 43 | 169.6 | 2.41 | 0.12 |
| BrS ~ s(HerrS) + s(Mack)* | 0.55 | 62 | 170.3 | 3.12 | 0.03 |
| BrS ~ s(HerrF) + s(HerrS) | 0.44 | 52 | 172.7 | 5.48 | 0.01 |
| BrS ~ s(HerrS)* | 0.41 | 52 | 177.6 | 10.36 | 0.00 |

## Prey biomass in marine ecosystem and breeding success

Two generalized additive models were nearly similar in explaining the relationship between northern gannet breeding success and the biomass of its main prey (Table 3). The model containing the smoothed function of fall herring and mackerel had a greater proportion of the variance explained ($R^2 = 0.70$) and deviance explained (DEV = 77%), with a similar AICc to the model containing only the smoothed function of mackerel biomass, but the smoothed function of fall herring biomass was not significant in the model ($P = 0.09$). Therefore, the most parsimonious model with the greatest explanatory power was the model containing only the smoothed function of mackerel biomass with a positive relationship (Fig. 2A, $\chi^2_{-2.3} = 1,068$, $P = 0.048$, $AICc = 165.9$, $delta = 0.00$, $weight = 0.54$).

## Diet of gannets and breeding success

The breeding success of gannets increased with the proportion of mackerel and herring in their diet. Years when the diet was dominated by these two species and therefore less diverse were beneficial for gannet reproduction (Figs. 2B–2D). For example, in 2005, the diet was dominated by mackerel (70%) and herring (28%) with a low Shannon H-index of 0.66 and a breeding success of 68%. In contrast, in 2019, the diet was very diversified (Shannon H-index = 1.73), with only 37% mackerel and a wide variety of species (*e.g.*, sand lance, redfish, capelin, squid, cod, etc.).

## Breeding success and divorce rate

The breeding success influenced the divorce rate with a lag of 1 year (Table 4, $F_{1,14} = 9.67$, $P = 0.008$), but not the inverse relationship, *i.e.* the divorce rate did not influence the breeding success at population level. These results suggest that changes in breeding success can be used to predict variation in divorce rate in gannets the subsequent year (Fig. 3). The most parsimonious model obtained for divorce rate (DivR) is presented in this equation (where 'BrS' is breeding success and '$t − 1$' is the year before divorce):

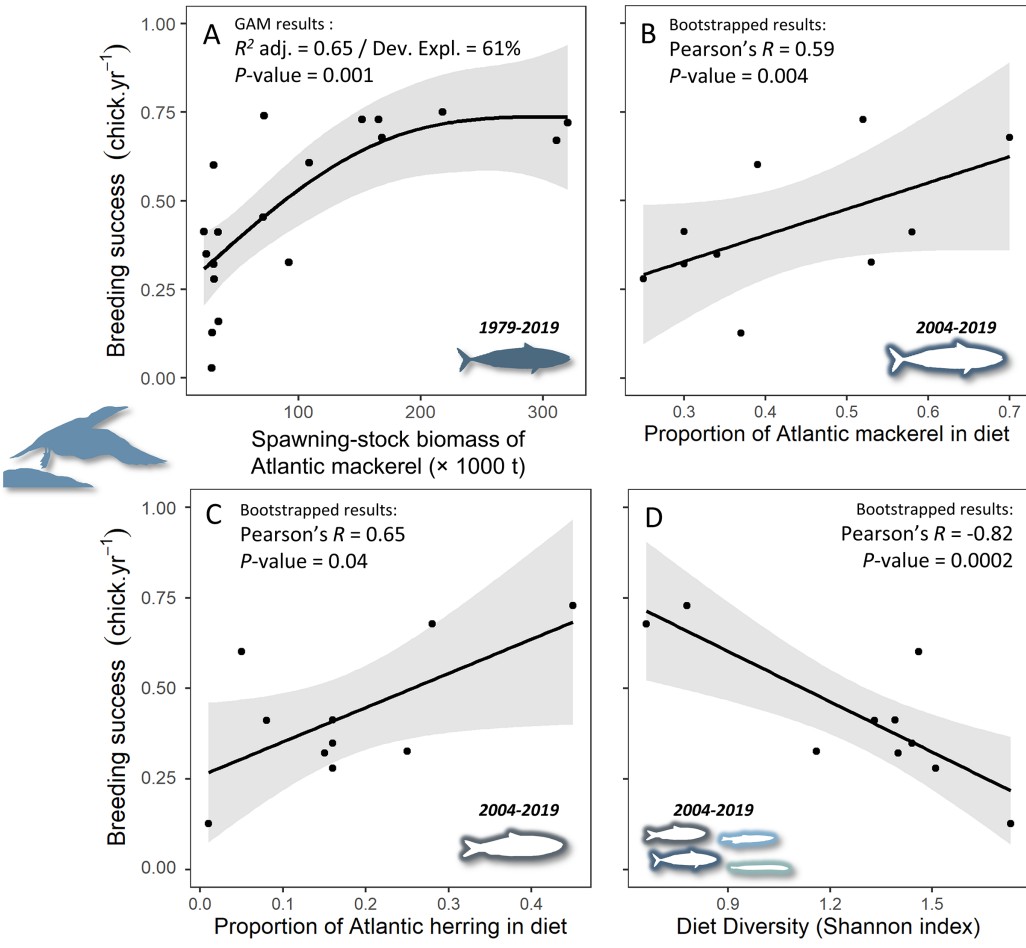

**Figure 2** Relationships between gannet breeding success and (A) biomass of Atlantic mackerel, (B) proportion of Atlantic mackerel in diet, (C) proportion of Atlantic herring in diet, and (D) diet diversity (according to the Shannon H-index calculation where the lowest values characterize a less diversified diet). Data from 1979 to 2009 were reported in *Guillemette et al. (2018)* and data from 2009 to 2019 were gathered by our team. Results showed in A were computed from a generalized additive model between the breeding success of northern gannets and the smoothed function of biomass of Atlantic mackerel (see Table 3). For B, C and D, a bootstrap significance testing approach was applied to estimate the *P*-value of the correlation coefficients.     

**Table 4** Results of the Granger-causality tests for pairwise comparisons of stationary time series for Northern gannet breeding success and divorce rate showing the *F*-statistic and *P* for one lag (expressed in years). The lagged term was determined empirically using the lowest value of *Akaike information criterion* (AIC) following comparison of models with a lag of 1, 2 and 3 years.

| | | Granger causality results | | | | |
|---|---|---|---|---|---|---|
| **Variable X** | **Variable Y** | **Lag** | **F** | **df1** | **df2** | ***P*-value** |
| Breeding success | Divorce rate | 1 | 9.67 | 1 | 14 | 0.008** |
| Divorce rate | Breeding success | 1 | 0.00 | 1 | 14 | 0.976 |

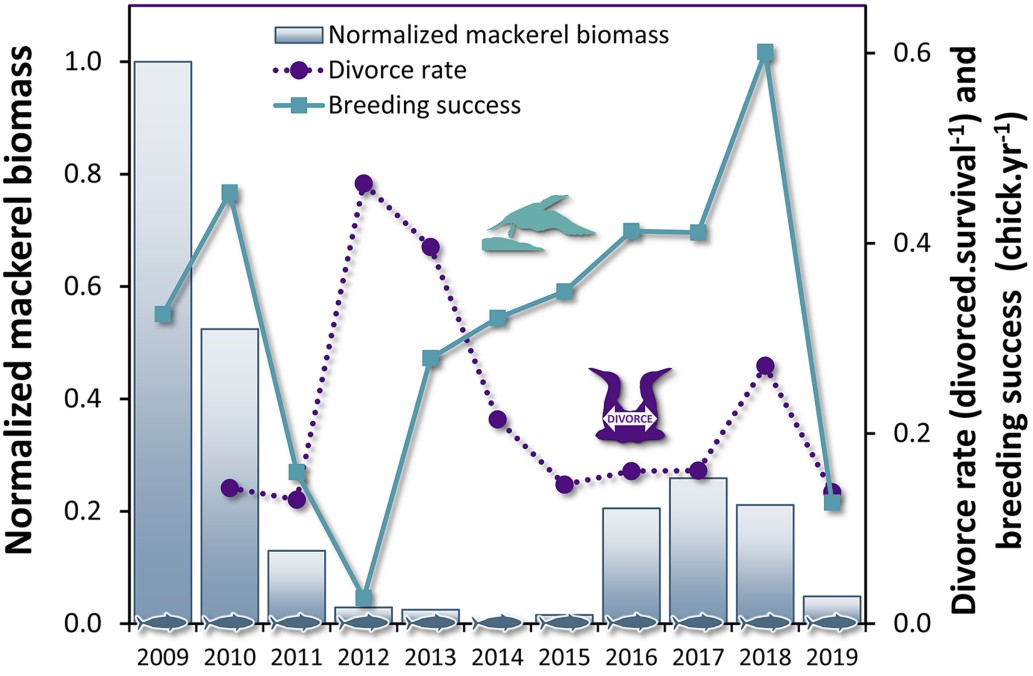

**Figure 3 Time series of divorce rate and breeding success in northern gannets, and normalized mackerel biomass in the Gulf of St. Lawrence between 2009 and 2019.**

$DivR_t = 0.41^{***} - 0.57^*\ BrS_{t-1}$ (***:$P < 0.001$, *: $P < 0.05$). In this equation, the significant coefficient $-0.57$ captures the cross-correlation between BrS and DivR 1 year later ($P = 0.01$). This model explains 89% variance in divorce rate (Adjusted $R^2 = 0.89$, $F_{2,8} = 43.38$, $P < 0.0001$).

## Annual differences in divorce rate

The proportion of gannets that divorced after breeding failure was almost three times greater overall than after breeding success (28% *vs.* 11%, Fig. 4). However, the probability that an individual divorce more after breeding failure than breeding success is highly variable across years according to best parsimonious model selected including the interactions between year $t$ and breeding success at year $t − 1$ ($\chi^2_{16} = 47.1$, $P < 0.001$, AICc = 850, delta = 0.00, weight = 0.974). Between 2010 and 2016, individuals experiencing breeding failure did not appear to be more likely to divorce the year after failure than individuals experiencing breeding success. However, the difference was significant between 2017 and 2019 ($P < 0.01$).

## Partnership status and breeding success

Breeding success in gannets was $0.32 ± 0.05$ chick.yr$^{-1}$ on average between 2009 and 2019 (Table 2). At the individual-level, the model with partnership status, time and interactions provided a significantly better fit to the data than the model with only partnership status, time or intercept alone (GLMM, $\chi^2_4 = 28.1$, $P < 0.0001$) (AIC = 2777, delta = 0.00,

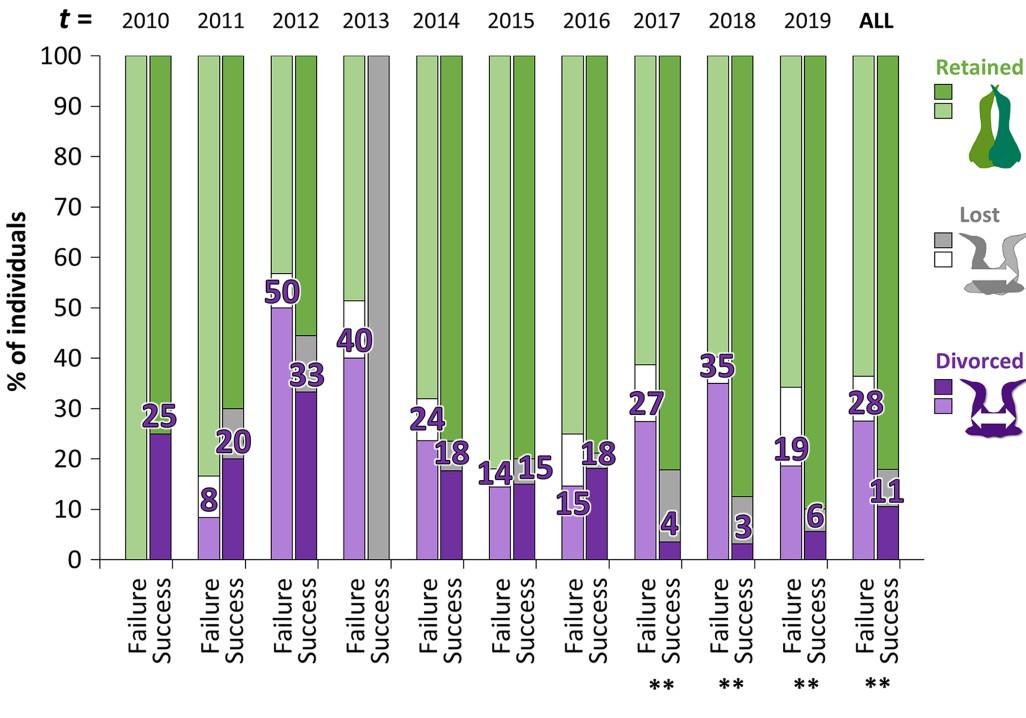

**Figure 4 Percentage of individuals that retained, lost their mate or divorced at year *t*, according to the breeding success at year *t* − 1 (with GLMM: divorce ~ breeding success at *t* − 1 × year *t* + (1|ID)).** For example, in birds that failed their breeding in 2017 (*t* − 1), 35% divorced in 2018 (*t*) whereas in birds that succeeded their breeding in 2017 (*t* − 1), 3% divorced in 2018. Pairwise comparisons were made for each year *t* between individuals that divorced after breeding failure or after breeding success with the "emmeans" package and significant differences are represented by asterisks (** = *P* < 0.01).

weight = 1.00). According to the area under the curve (AUC) that measures performance of the model (where AUC = 1 is perfect), the discrimination power of the full model is very good (AUC score = 0.83). Breeding success values and interannual trends were different according to the partnership status and time (Fig. 5). At year *t*, when partnership status is determined for an individual, breeding success was significantly different between partnership categories. Individuals that retained their previous mate presented higher breeding success (0.35 ± 0.02 chick.yr$^{-1}$), followed by divorced birds (0.24 ± 0.03 chick.yr$^{-1}$) and individual that lost their mate (0.16 ± 0.04 chick.yr$^{-1}$) the previous year (*t* − 1). There was no significant difference between 'divorced' and 'lost' categories at year *t* (*P* = 0.24). At year *t* − 1, retained birds had a two-fold higher breeding success than divorced birds (*P* < 0.0001). In each partnership status categories, pairwise comparison indicated different trends. Gannets that retained their partner decreased their breeding success at year *t* (*P* = 0.03) and at *t* + 1 (*P* = 0.002). At the opposite, gannets that divorced at year *t* doubled their breeding success after mate change from *t* − 1 to *t* + 1 (*P* = 0.04). One year after the partnership status determination, breeding success was similar between all individuals (*P* > 0.97).

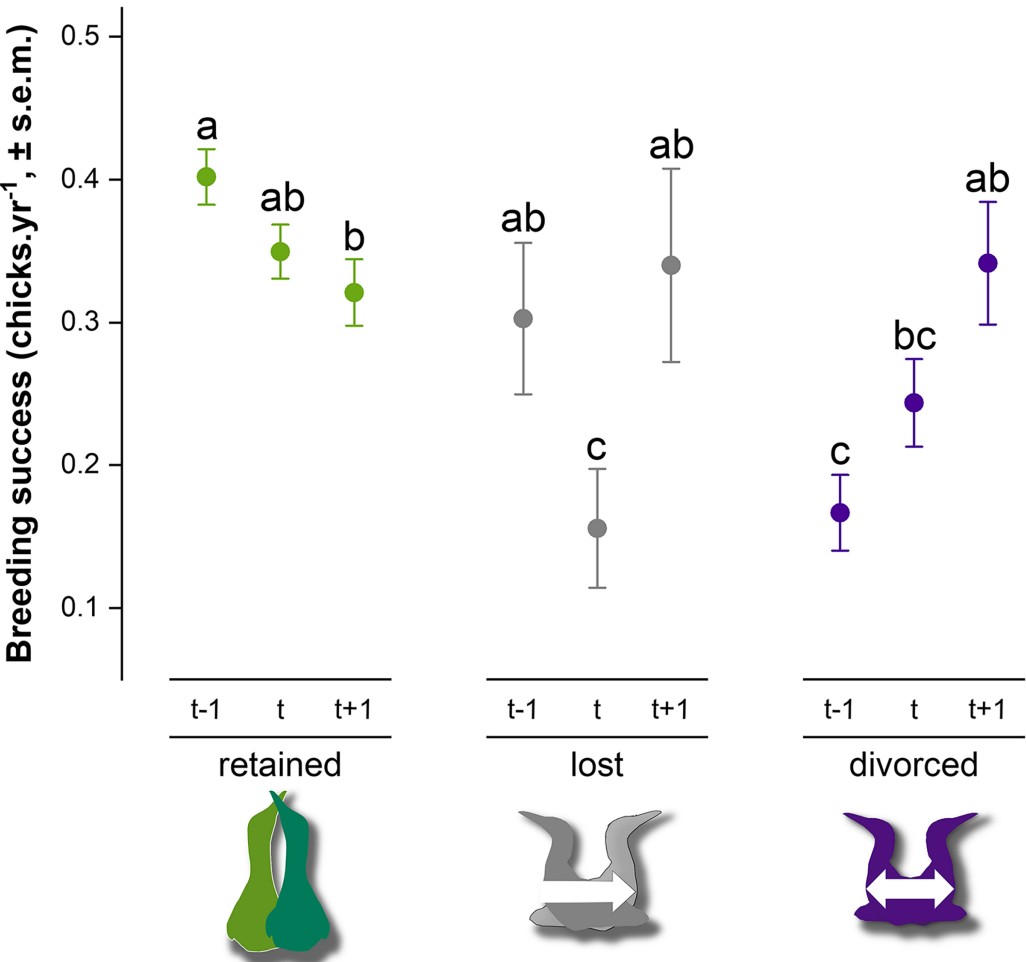

**Figure 5 Breeding success comparison between partnership status of northern gannets and time, where $t$ is the year of the partnership status determination, $t − 1$ is the previous year, and $t + 1$ is 1 year later.** A generalized linear mixed model (GLMM) was used to test differences in annual breeding success between partnership status and time (with binomial distribution, link = "logit"). Pairwise comparisons were performed using estimated marginal means and *post hoc* pairwise comparisons adjusted by Tukey were applied to test group differences. Similar letters mean that there is no significant difference between the categories ($n_{retained} = 608$, $n_{lost} = 52$, $n_{divorced} = 123$).

## DISCUSSION

Despite global changes in marine ecosystems, seabirds show behavioral resilience during the breeding season. As we showed that divorce was triggered by breeding failure and ensured improved individual short-term breeding performance, our results strongly suggest that divorce acts as an adaptive strategy in northern gannets. This behavioral flexibility would be driven by a 'success-stay/failure-leave' pattern (*Schmidt, 2004*) that is mainly influenced by the abundance of its preferred prey in the marine ecosystem of eastern Canada: Atlantic mackerel. The abundance of this pelagic fish was at its lowest in the last decade, as was the breeding success of gannets. However, during this period, we observed that years of breeding failure were followed by an increase in divorce rate. After

breeding failure, gannets were more likely to change partners. Such behavioral flexibility observed during reproduction is a good example of resilience in a monogamous seabird species under food uncertainty.

## When do gannets divorce?

Our results are consistent with our first hypothesis: divorce is a result of breeding failure and prey decrease. As predicted, divorce is more often observed in northern gannets 1 year after breeding failure while breeding failure is induced by mackerel decrease in the Gulf of St. Lawrence. *Guillemette et al. (2018)* showed that breeding success of gannets between 1979 and 2014 starts to decline below a threshold of mackerel spawning-stock (mackerels over 3 years old) biomass of 132,300 t (or 97,370 t when considering biomass corrected for size of fish (<35 cm) available to the gannets). During the period in which we studied the relationship between breeding success and divorce rate (2009 to 2019), even if we used the total stock biomass of mackerel (including mackerel of 0 to 10 years old), the biomass was quite below this threshold on average (67,336 ± 8,623 t, except for 2009: 141,243 t). These results illustrate that mackerel abundance was at its lowest during our study period. With the analysis of the diet during the same last years, the observed relationship between proportion of this prey and breeding success was similar. We also observed such a relationship with Atlantic herring in the diet, but it is almost three times less important than mackerel. The decrease in the presence of mackerel in the diet also results in an increase in diet heterogeneity, with a negative impact on breeding success. It further supports the importance of mackerel in the diet of gannets during the chick-rearing period (*Garthe et al., 2007*; *Guillemette et al., 2018*) as an extrinsic modulator of breeding performance of gannets in the Gulf of St. Lawrence.

Consequently, under conditions of low availability of their preferred prey, both parents increase their time spent foraging (*Guillemette et al., 2018*) and leave their offspring temporarily unattended. Normally, seabirds divide parental care between sexes: one parent leaves for foraging and the other waits for its return by protecting its chick (*Schreiber & Burger, 2002*). However, when food is scarce, foraging trips are longer and further away from the colony, inducing the attendant partner to leave the nest to ensure its own survival, leaving a younger chick or the chick earlier (*Regehr & Montevecchi, 1997*). The unattended chick becomes vulnerable to the assault of adults from neighboring sites (*Ashbrook et al., 2008*), attacked by adults from nearby sites (*Ashbrook et al., 2008*), or assaulted by non-breeders attempting to usurp sites (*Porter, Anderson & Ferree, 2004*; *Hamer et al., 2007*), or killed by predators (*Oro & Furness, 2002*), which turns out very often into breeding failure. As observed in other monogamous bird species, a poor breeding performance recorded at a specific nest site with a specific partner influences negatively the occurrences of pair reunion at this site (*Ens, Choudhury & Black, 1996*; *Bried & Jouventin, 2002*; *Dubois & Cézilly, 2002* but see *Choudhury, 1995*; *Taborsky & Taborsky, 1999*).

However, according to a review of 93 species, seabirds exhibit generally very high fidelity to their nesting site and their partner, 75% and 82% on average, respectively (*Schreiber & Burger, 2002*). In this review, it is indicated that northern gannets have very high nest and mate fidelity (90% and 83.5% respectively), compared to other species.
However, divorce rate estimates reported for two closely related species vary between 40–43% for Australasian gannet (*Morus serrator*) (*Ismar et al., 2010*), and 45% for blue-faced booby (*Sula dactylatra*) (*Kepler, 1969*), which are twice as high as the results obtained for northern gannets in our study. The only divorce rate reported in literature for northern gannets (17% in *Cézilly, Dubois & Pagel (2000)* from *Nelson (2002)* at Bass Rock Island, Scotland, between 1961 and 1976) is probably an overestimate of divorce rate because the methodology described included the loss rate. According to our study, the mate change rate (including loss and divorce rates) was on average 30 ± 14% (where loss rate was 7 ± 4% (range: 4–13%) and divorce rate, 22 ± 12%). Considering that loss rate is dependent of adult survival rate (less influenced by food conditions during breeding period) and assuming that the loss rate measured in our study may be used as a fair estimate, we may expect that the more exact divorce rate reported in *Cézilly, Dubois & Pagel (2000)* should be between 4% and 13% (mean ± SD: 9 ± 4%) between 1961 and 1976. Breeding success recorded during the sixties and the seventies was very high and ranged between 73% and 85% at Bass Rock colony. In this context, divorce rate recorded in a period of poor food conditions at Bonaventure Island (between 2009 and 2019) would be around 2.6-fold higher than the rate of divorce observed during better environmental conditions. During the seventies, in the Gulf of St. Lawrence, mackerel biomass was widely higher than now (mean ± SD: 235,313 ± 75,950 t, *Smith et al., 2020*). Thus, we could infer that divorce rate was lower during this decade (and after, during the eighties and the nineties).

Breeding habitat quality and environmental predictability have been proposed for some time as factors influencing temporal variability in divorce rate. As an example, European blackbirds (*Turdus merula*) increase their divorce rate when they are in low quality nesting sites (*Desrochers & Magrath, 1993*). Recently, a study on a long-lived seabird species (black-browed albatross, *Thalassarche melanophris*) has demonstrated for the first time empirical evidence that the prevalence of divorce can be directly modulated by environmental temporal variability *via* sea surface temperature variations (*Ventura et al., 2021*). In lower quality years, with warmer sea surface temperature anomalies, the probability of switching mate increased in albatross populations. The underlying mechanism proposed to explain the link between sea surface temperature and divorce rate is through a bottom-up process of reduced food availability during warming periods (*Behrenfeld et al., 2006*), causing a subsequent reduction in breeding success. Thus, our results support the proposed explanatory hypothesis directly linking food abundance, breeding success and divorce rate.

However, our study does not allow us to identify the temporal mechanism of pair formation and mate change at fine scale (timing and proximal causes of divorce) but we explore here few aspects guided by our results. Timing of the formation of old and new pairs, or their break-up, is different between migratory and resident species and between species with continuous and part-time partnership (*Ens, Choudhury & Black, 1996*). Northern gannet is a migrant species but it is unknown if individuals migrate in pairs or if they reunite on their winter areas as waterfowl species do (*Robertson & Cooke, 1999*). Pair bonding behavior is unknown during winter for gannets. In two studies reporting

migration destination for both sexes in four couples (*Fifield et al., 2014*; *Pelletier et al., 2020*), partners wintering in the same area were reported in only one couple (*Pelletier et al., 2020*). Sex differences in migration patterns are common in seabirds (*Phillips et al., 2009*; *Bosman et al., 2012*), and males are generally wintering closer to the colony than females (*Phillips et al., 2017*). The same is observed for northern gannets nesting in Europe (*Deakin et al., 2019*), but both sexes are observed throughout the species' winter range of gannets nesting in North America (*Fifield et al., 2014*; *Pelletier et al., 2020*). Because no difference is detected in the arrival date at colony between males and females (*Pelletier et al., 2020*), it suggests that gannets nesting on Bonaventure Island reunite, form new pairs or break-up from their partners essentially on the breeding ground. We thus hypothesize that individual's decision process to divorce probably occurs at the colony at the beginning of breeding season.

## Why do gannets divorce?

At the individual level, gannets brooding a chick until fledging tend to stay with the same partner while gannets that have failed breeding are more likely to divorce. In a context of food depletion of the preferred prey as observed between 2009 and 2019 in the Gulf of St. Lawrence, individuals that change partners seems to optimize their fitness (1 year after the divorce). It supports the hypothesis that divorce may be an adaptive strategy as it would not happen by chance. Divorce in gannets may be a form of adaptive mate choice, triggered by low breeding success (*Dubois & Cézilly, 2002*) and by the potential to improve their breeding success with a new partner (*Choudhury, 1995*; *Black, 1996*). During a period of low food abundance, gannets that divorce seem to benefit from this change since they increase their breeding success the subsequent years with the new partner. Our results reflect the "win-stay/lose-switch" theory (*Switzer, 1993*; *Naves, Yves Monnat & Cam, 2006*; *Piper, 2011*), which hypothesizes that the decision to repeat a breeding event on the same territory or with the same partner will depend on past performances. Other studies support the theory demonstrating that previous breeding failure predicts infidelity (*e.g.*, *Bai & Severinghaus, 2012*; *López-López, 2016*), but it is not always the case. For example, prior reproductive success was not predictive of divorce in Australasian gannet (*Morus serrator*) (*Ismar et al., 2010*). These conflicting results reveal the complexity of the mechanisms underlying mate fidelity and suggest that this behavioral modification induces variable physiological costs depending on individual quality and ability to respond to stressors.

According to literature reviews written on divorce in monogamous birds (*Choudhury, 1995*; *Culina, Radersma & Sheldon, 2015*), our results suggest that divorce is a short-term adaptive strategy to counteract breeding failure occurring when birds are faced with reduced food supplies. Therefore, divorce should be viewed as a reproductive strategy that maximize individual fitness (*e.g.*, *Coulson, 1966*; *Ens, Safriel & Harris, 1993*). According to various hypotheses reviewed by *Choudhury (1995)*, it is difficult and hazardous to determine which is the best hypothesis that explains the divorce in a specific bird species. For instance, gannets that divorce for improving reproductive success may both change partners because their combined qualities result in reduced fitness (incompatibility hypothesis, *Coulson & Thomas, 1980*), or one member of a pair changes to improve its

reproductive success by obtaining a better-quality mate (better option hypothesis, *Ens, Safriel & Harris, 1993*), or divorce may arise from errors made in the original choice of a mate (errors of mate choice hypothesis, *Johnston & Ryder, 1987*). These hypotheses are not mutually exclusive and only an experimental study in which the conditions and quality of the partners are controlled would reveal the one that would best explain the divorce in northern gannets.

Our study showed that divorce can be seen as an example of behavioral flexibility to counteract the low productivity observed during a period of limited food resources. Indeed, temporal fluctuations in colony productivity were directly related to Atlantic mackerel abundance and inversely related to divorce rate. At the individual level, gannets that change partners do so following a reproductive failure and there is an increase in reproductive success 1 year following the divorce. In the context of rapid environmental changes (accelerated by anthropogenic pressures), behavioral flexibility would be important because opportunities for dispersal and adaptation are often limited for seabirds. These behavioral responses can therefore lead to the avoidance of ecological traps in which the demographic parameters of the animals (as birth, death, and migration rates) could be altered. Gannets that divorce could improve their individual fitness by increasing their subsequent breeding success. However, breeding success is only one component of fitness and divorce could also affect survival (*Nicolai et al., 2012*; *Culina et al., 2013*) and probably decrease lifespan. In a future study, the cost and benefits of mate choice in northern gannets should be explored at physiological and behavioral levels to understand and explain the potential short- and long-term consequences of an individual's mating decision process on stress regulation and individual health status.

## ACKNOWLEDGEMENTS

We acknowledge the contribution of the long-term monitoring programs of mackerel and herring abundance data, fishing statistics, and commercial samples from the Department of Fisheries and Oceans (DFO) Canada (Québec region). Thanks for the extensive commitment of sampling, laboratory and analytical personnel involved in the Laboratoire d'ornithologie marine de Rimouski (LOMR) at Université du Québec à Rimouski (Yannick Seyer, Laurie Maynard, Sarah Wing, Isabeau Pratte, Mélanie Laflèche, Catherine Ayotte, Dévrig Bouillet, Félix Larochelle, Olivier Buteau, Gabrielle Théroux, Liette Régimbald, Catherine Gloutnez, Sandrine Gingras, Jolanie Roy, Safouane Khamassi, Jeanne Bouchard, Selma Elfassi-Fihri, Marie-Anne Robitaille, Richard Gravel, Roxanne Turgeon, Camille Novales, Laurence Gagnon, Marie-Eve Labonté Dupras, Catherine Destrempes, Catherine Bouchard, Fanny May Couture-Charron, Laury-Ann Dumoulin, Angéline Robichaud), Cégep de Rimouski (Gabrielle Bouchard, Anne-Charlotte Lebel, Daisy Turcotte, Laura Turcotte, Andréa Lévesque, Isabelle Demalsy, Alexia Tremblay, Emma Côté, Lydiane Parent, Sandrine St-Pierre-Lepage), CWS and DFO during all those years (particularly Jean-François Rail, Andrew Smith, Elisabeth Van Beveren, and François Turcotte). We sincerely thank France Dufresne (UQAR) for having welcomed us in its laboratory for the sex determination analyzes and to Geneviève Côté, Jean-Michel Martin and Frédérique Paquin for helping us. Thanks to Parc national de l'Île-Bonaventure-et-du-Rocher-Percé

and its staff for transportation and supportive collaboration. Final thanks to André Desrochers and many anonymous reviewers for helpful comments. The title is inspired by "The Times They Are a-Changin" which of course refers to one of the most powerful songs written and composed by Bob Dylan.

### Funding

This work was supported by the Canadian Natural Sciences and Engineering Research Council (NSERC) discovery and equipment grants to Magella Guillemette, by the Fonds de recherche du Québec-Nature et technologies (FRQNT) Research program for college researchers to David Pelletier, and by the NSERC Alexander Graham Bell Canada Graduate Scholarship to David Pelletier. The funders had no role in study design, data collection and analysis, decision to publish, or preparation of the manuscript.

### Grant Disclosures

The following grant information was disclosed by the authors:
Canadian Natural Sciences and Engineering Research Council (NSERC).
Fonds de Recherche du Québec-Nature et Technologies (FRQNT).
NSERC Alexander Graham Bell Canada Graduate.

### Competing Interests

The authors declare that they have no competing interests.

### Author Contributions

- David Pelletier conceived and designed the experiments, performed the experiments, analyzed the data, prepared figures and/or tables, authored or reviewed drafts of the paper, and approved the final draft.
- Magella Guillemette conceived and designed the experiments, authored or reviewed drafts of the paper, and approved the final draft.

### Animal Ethics

The following information was supplied relating to ethical approvals (*i.e.*, approving body and any reference numbers):

All bird capture and handling methods were approved by the Animal Care Committee (ACC) of the Université du Québec à Rimouski (CPA-49-12-102, CPA-65-16-177), and complied with the guidelines of the Canadian Council on Animal Care (CCAC).

### Field Study Permissions

The following information was supplied relating to field study approvals (*i.e.*, approving body and any reference numbers):

Field experiments were approved by Canadian Wildlife Service-Environment and Climate Change Canada (permit numbers SC25, RE-27) and by Société des établissements de plein air du Québec (permit numbers PNIBRP-2008-001 to PNIBRP-2019-001). Birds

were marked with a U.S. Fish and Wildlife Service steel ring and an alphanumeric coded and colored plastic band (permit number 10704).

## Data Availability

The R scripts and raw data are available in the Supplemental Files.

## Supplemental Information

Supplemental information for this article can be found online at http://dx.doi.org/10.7717/peerj.13073#supplemental-information.

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
