# Peer review of "Times and partners are a-changin’: relationships between declining food abundance, breeding success, and divorce in a monogamous seabird species"

_PeerJ, doi:10.7717/peerj.13073_

## Round 0.1 · original submission · Major Revisions

The three reviewers were positive regarding the overall approach, but they had concerns regarding the statistical analyses and the use of "Granger causality". I share these concerns, and I would ask you to carefully consider the comments made by the reviewers. I have used myself time series analyses, and there are some aspects that I find surprising, like retaining Div with one year lag (your equation 1) whereas the P-value for that coefficient is 0.57. AIC is known to be a bit liberal (that is the best model may include coefficients that are not "significant" at the 0.05 level), but not as much as that. I would also like to see scatter plots showing the selected relationships to be able to assess the linearity of the relationships, and the occurrence of outliers. Generally, I would advise that you rely less on statistical significance and more on relationships that make sense biologically as pointed out by reviewers. 11 years represent a lot of field effort, but statistically when trying to fit models with different covariates at various lags, it is a very small sample size and therefore open to high level of variability (including P-values).

Divorce rate is a proportion, and should in principle be analysed using a binomial (Bernoulli to be precise as it is 0/1 at the individual level) distribution for the response,as you do with breeding success. It makes it difficult to use classical time series models as they often assume a normal distribution with constant variance for the residuals, but you need at least for the most important relationship (mackerel-divorce) to show that results are unaffected when you account for the different sample sizes in different years. See e.g. Mercier et al. 2021 (which is also relevant as kittiwakes had rather high divorce rates too).

l. 256: you write that "Breeding success in gannets was 32 ± 16% in average between 2009 and 2019 (Table 2)", but in Table 2, 16% corresponds to SD, not SEM as written l. 230. Please check that you are consistent.

In the reference list, there are some typos (eg Sula bassana appears as sula bassana a few times), please check the list carefully.

Mercier, G., Yoccoz, N. G., & Descamps, S. Influence of reproductive output on divorce rates in polar seabirds. Ecology and Evolution doi:https://doi.org/10.1002/ece3.7775

Reviewer 1 ·

Basic reporting

There are minor grammatical errors which I have tried to address

Experimental design

No comment

Validity of the findings

There are a few parts of the discussion that would benefit from being expanded to widen the scope.

Additional comments

I really enjoyed reading this article. I think it is novel and interesting and well analysed and reported. While it looks like a lot of comments they are minor edits to the grammar, suggested rephrasing and request for clarifications
Title – I am definitely not a fan of puns in titles but I appreciate why some people are. However, here comes the sun is in no way related to your work so I suggest you remove it or think of a more suitable replacement
Line 12 – this first sentence is very, very broad and a bit vague. I think you could make this a bit more focused on life history perhaps? Or reproduction?
Line 14 – I would rephrase to read “ at the population and individual level”
Line 19-20 – I would have thought this should be in the past tense?
Line 22 – is this always the case? If not how often would be helpful here
Line 30 – resilient to environmental change?
Line 32 – I would rephrase to be “Normally seabirds divide parental care between the pair” or something like this? It is a bit clunky as it is and is missing an apostrophe
Line 33 – why must they foraging simultaneously when there is food depletion? If there are on an egg or a chick they can not? I don’t really understand this sentence. They can leave the offspring unattended once it reaches a certain age? I can’t imagine they would leave a tiny chick or an egg? Maybe you could rephrase this to cover the idea that when food is scarce they are more likely to leave a younger chick or the chick earlier to obtain sufficient food?
Line 36 -I would state that this is for colonial breeders – or birds that nest in tight colonies. If they are in loose colonies or burrow nesters this would not be true.
Line 40 – why would this affect pair reunion? Because failure would do? I think it is the word therefore that is a bit confusing here. I would just say that this reduction in breeding performance can then affect pair reunion
Line 44 – you say 80% of birds are monogamous then they have diverse mating systems? Isn’t this contradictory? Do you mean the extent of true monogamy varies because 80% of spp are socially monogamous?
Line 47 – given you look at individual level variation would this be a good place to mention this variation too?
Line 50 – so do you include widowhood in divorce? By saying even it suggests you do but true divorce does not include instances where one partner dies
Line 56 – you can have biparental care without social monogamy….maybe “and the associated social monogamy”
Line 62 – this is quite vague – a random event. What about if a better partner comes along? Birds can upgrade their partner even if they haven’t failed. For example for a younger partner or one in better condition. I think you be a bit more thorough in this part.
Line 65 - but not only conditions right?
Line 67 – but this is true for life time reproductive success but for a time lag of 1 this is a little simplistic.
Line 95 – typo
Line 97 – why are you repeating why gannets divorce?
Line 115 – was this from observations?
Line 173 – what does number of complete pairs monitored mean? Is this the pair retention? That would seem very low
Line 189 – delete does
Line 199 – for bivariate time series comparisons – do you mean pairwise comparisons between two variables? For people less familiar with these techniques an extra line here could be helpful
Line 223 – “individual was”
Line 233 –“showed substantial”
Line 237 – “were very different”
Line 239 – “was” , “accounted” – needs to be past tense throughout the results
Line 243 – I would personally drop “VAR model diagnostics” as this is in the methods
- What does it mean given the lack of an inverse relationship? This would also be good to state
- I would redefine the variables where the equation is shown
- I think it would be better not to use abbreviations in table 2
Line 247 – is it worth including these results here or explain them a bit here?
Line 262 – 263 – this sentence doesn’t really make sense - rephrase?
Line 266 – what is a lost category?
Line 249 – I don’t think therefore is the right word here. Is this sentence describing the pattern from the result in the one before? If so then I would say “these analyses showed” or something explicit
Line 252 – what is an impulse? I would explain again here
Line 257 – delete does
Line 258 - delete both the
Line 281 – I would reiterate the hypothesis here
Line 282 – longer than what
Line 283 – biomass of mackeral?
Line 287 – biomass
Line 288 – has shown….were related
Line 293 – see comments in the intro – they always leave it unattended eventually so I think you need to be more specific here
Line 300 -this review
Line 301 – Otherwise should be However…estimates of divorce rates
Line 302 – vary
Line 312 - figured is not the right word here
Line 314 – the rate of divorce – I don’t think that occurance is the figure given here
Line 316 – actually? Do you mean than now?
Line 317 – I would rephrase as “High concentrations of DDT were another extrinsic factor….”
Line 322 – does not permit “us” to …
Line 324 – should be present tense
Line 326 – part-time partnerships is not a standard term – do you mean
low monogamy? Or birds that are only together when they breed and not during winter?
Line 326 – Northern gannets are a migratory species … (it is not obvious if you don’t know about gannets!)
Line 329 – but do waterfowl go onto land in the winter? If gannets stay at sea all winter is it likely that they meet their partners at sea? Seems unlikely
- What does one and three pairs mean?
Line 321- Why would selection favour the effort of finding you partner at sea?
Line 335 – there is no difference detected…
- This whole sentence is a bit muddled – why does leaving at the same time and females coming back later suggest the same thing? The final sentence could be better phrased – do you mean that decisions on divorce occur at the breeding ground?
Line 356 – change succeed to support to just support
Line 359 – here you show in another very closely related spp that divorce does not follow breeding failure. I think in the discussion it would be good to explore why this may be in some seabirds. I think in very long lived spp where pair experience predicts fitness the cost of divorce is high and so you would expect divorce to come at a higher cost. In order to extend the findings beyond that of this population, I think it would be nice to explore why the patterns are so mixed in seabirds. I also think it would be good to chat about how predictable or high quality the environment – if there is just one bad year would divorce be adaptive?

Reviewer 2 ·

Basic reporting

In general, I found the manuscript entitled “Here comes the sun: divorce in a seabird species increases breeding success during low food abundance” to be an interesting examination of how a widely-accepted concept (low breeding success leads to divorce in long-lived seabirds) operates in practice. In particular, I appreciated the extension of the analysis to include data on prey abundance, allowing for a more complete picture of how the lack of prey availability may directly lead to breeding failure, and, by extension, mate switching. I did have several thoughts about the analysis and the narrative, which I outline here, but I think that after revisions the study will make a worthwhile contribution to the literature on the topic. The style and grammar of the written text could be improved a bit by some copy-editing, but it was understandable throughout.

Experimental design

There was one aspect of the analysis caused me to think quite a bit, and I’m not sure I ultimately came up with a good answer: each of the time series fed into the Granger-causality tests was first detrended. I am not very familiar with Granger-causality, but that seemed like an appropriate step, because of the requirement that each series be stationary. However, as I considered an individual gannet’s mate choice from a biological perspective, I wasn’t sure if a bird would be capable of reacting to detrended prey or breeding success patterns—would not a gannet’s breeding success to react to the absolute prey abundance, and then a mate choice decision to react to the actual breeding success in prior years? However, I do see the value in removing any existing long-term trends, which could be the result of many other factors: for example, both breeding success and divorce rate could be decreasing as a result of some third, unknown factor also subjected to long-term trends. So, I found myself doubting whether an analysis that relies on detrended time series would be appropriate for this question. I wonder if the authors could help explain (either in the methods or discussion) exactly why the process of detrending the data is necessary, and perhaps discuss the ramifications of doing so in the interpretation of the results.

I may be mistaken, but I assume that the lag of 2-years found between mackerel SSB and breeding success (Table 3) should be interpreted such that breeding success is lower 2 years after a drop in mackerel stocks, correct? If so, I found that to be very counter-intuitive given my naked-eye interpretation of Fig 2, in which the most obvious pattern appears to be that breeding success dramatically declines in the year immediately following major declines in mackerel SSB. I suspect that a stronger 2-year lag may be the result of running the analyses on detrended data, and perhaps even a change in the feeding ecology of the gannets after the apparent crash in mackerel stocks in 2011. The authors do discuss the gannet diet briefly in the discussion, but any information about how breeding success increased so dramatically following the dramatic decline in 2012 would be helpful. Given that the time series seems to be so dominated by a large crash in mackerel abundance in the early period, and then a disassociation between mackerel and breeding success in the later period, is the Granger-test approach well-suited to these data? Some explanation for that counter-intuitive result may be useful for the reader in the results or discussion.

Validity of the findings

I found the individual results and model of partnership status, including Fig 3, to be the most compelling part of the argument. Those results told a very clear story about the difference in repairing after partner-loss and divorce. I also found the separation of the discussion into the “when” and “why” sections to be very effective.

Additional comments

My main suggestions for revisions are to further examine the ramifications of some of the methods associated with the Granger-causality tests, as described above, and to explain why those results seem to provide a counterintuitive lag between mackerel SSB and breeding success. Without more expertise on the Granger approach I cannot say whether changes to the analytical method are warranted, but I do feel that there should at least be some additional explanation for other readers who, like me, are unfamiliar with such methods.

Some minor edits to consider as well:

L108: Is it possible that the focus on peripheral areas of the colony influenced the rate of divorce? It is often suggested (for some species at least) that young/inexperienced birds may inhabit these areas, where breeding success is lower than near the center of a colony. That being said, I believe that actual empirical evidence for it may be quite rare (though, perhaps some work on Adélie Penguins may have discussed it?) If so, it is possible that the breeding success and rate of divorce observed here may not be representative of the entire colony.

The paragraph beginning on L79 seems to be making several loosely-related points about gannet biology in the region, and it is difficult to follow the narrative thread. A reorganization of the paragraph, perhaps splitting it into one paragraph on feeding ecology/prey and another paragraph on why it is a good model for studying divorce, may help.

L121: Does the section on sex determination need that much detail? Perhaps simply citing Griffiths et al. 1998 allows the rest of the paragraph to be much shorter? On the other hand, these details may be useful to some readers, so it may be better to keep them in, as long as the manuscript is not too long.

During the description of the Granger-causality methods, there were several terms used (such as “stationary”, “impulse”, and “shocks”) that may hold different connotations for most seabird biologists. In general, the section was fairly good at explaining the methods to a lay audience, but some additional explanation for the concepts in L194-210 may be helpful.

L211: It was not clear to be at first what “annual means” referred to, and why these methods were being employed. Perhaps an additional sentence at the beginning of this paragraph explaining how these methods connect to the rest of the methods would help.

Equation 1: I was unsure of the purpose of the various asterisks.

L374: The conclusions section isn’t really a summary, as it adds more relevant discussion—I would simply remove the ‘Conclusions’ header, or perhaps move some of the material out of this section and into another part of the discussion.

Reviewer 3 ·

Basic reporting

This is a really interesting premise and a clear gap in the literature. The authors have identified a colony with comparably high divorce rates to the broader biogeographical range and use this to study the causes and consequences of divorce in the framework of an adaptive strategy. The paper is generally well written. There were some sentences that were hard to follow and others where parts seemed to be juxtaposed, and I have highlighted/corrected these in the attached annotated PDF.

Figures were very well presented and easy to follow, with really nice iconography to aid interpretation about pairs that stayed together and divorced. I did find the associated data difficult to use to replicate findings. Divorce rate data in particular did not identify the years that observations occurred in (just year t, t-1, t+1 etc). While that could be used with the provided code to run an analysis, there is no way of checking the underlying data to understand which years corresponded to poor food or breeding success. Perhaps providing raw data here would be a good idea, identifying nests and partners?

Experimental design

This is clearly original research that falls within the remit of the journal. The research questions could be better defined, for example, whether you expect a positive or negative relationship between variables rather than just a relationship.
Methods could do with some clarification to aid understanding and replication, but it is generally easy to follow and quite obvious that a significant amount of work has gone into field data collection for which the authors should be rightly proud.
Unfortunately I am unfamiliar with the analysis approach taken (I'm far more familiar with the use of Hidden Markov models for time series analysis that can be correlated with environmental variables), so can't really comment on the appropriateness of this. from the methods, it seems to be appropriate in terms of accounting for temporal autocorrelation and identifying time lags, but I can't really comment on the mechanics of the analysis.

Validity of the findings

this is probably where I had the greatest issue. While it is a very interesting study and there are clear results, I'm not convinced that the results are properly reflected in the paper title or conclusions. The paper suggests that poor food availability with a season leads to low breeding success and that this is caused by both parents having to forage and chicks being killed by aggressive neighbours. It ignores the fact that if both parents need to forage, that they may be maintaining their own body condition and chicks are simply not getting enough food to meet requirements. Perhaps the authors could support this with information on the weights or condition of chicks found dead in the colony?
Following poor breeding success as a result of low food availability, pairs are more likely to divorce. While they did show a convincing 1 year time lag between breeding success and divorce, there was no relationship between food and breeding success within seasons, in fact there was a 2 year time lag that seems to have been interpreted as demonstrating a link between food and breeding success that just doesn't make biological sense. I'm also unsure how the model accounted for the apparent increase in food conditions following a poor year, particularly for those pairs that remained together following a poor food season and breeding failure. Obviously if nests fail due to poor food availability in one year, if conditions are better the following year, breeding success should be higher across the board and not just on those pairs that divorced? And this may differ to those birds that were successful in the poor food year, but possibly unsuccessful the following year - perhaps there is some carry-over effect? I was expecting a basic comparison of breeding success between failed nests that stayed together and failed nests that divorced in the year following breeding failure to disentangle increased food availability in years following divorce from the effect of divorce. This may be where my lack of expertise in the statistical approach lets me down, but I imagine other readers would have similar questions, so if this is accounted for in the model, perhaps this needs to be better explained.

Additional comments

I have provided a number of additional comments etc on the attached annotated manuscript PDF.

Annotated reviews are not available for download in order to protect the identity of reviewers who chose to remain anonymous.

---

## Round 0.2 · accepted · Accept

The reviewer appreciated the work done for the revision and I agree. I also appreciate all the datasets and scripts (the only issue i found was the name of the files - PeerJ is adding peerj-64345- to the names, but that should be easy to correct by users).

Reviewer 1 ·

Basic reporting

The authors have done a great job addressing my comments and I have no further comments

Experimental design

The authors have done a great job addressing my comments and I have no further comments

Validity of the findings

The authors have done a great job addressing my comments and I have no further comments

Additional comments

The authors have done a great job addressing my comments and I have no further comments